# Immunohistochemical Femoral Nerve Study Following Bisphosphonates Administration

**DOI:** 10.3390/medicina56030140

**Published:** 2020-03-19

**Authors:** Vasileios Alexandros Karakousis, Danai Liouliou, Aikaterini Loula, Nikoleta Kagianni, Eva-Maria Dietrich, Soultana Meditskou, Antonia Sioga, Theodora Papamitsou

**Affiliations:** 1Laboratory of Histology and Embryology, Aristotle University of Thessaloniki, 54124 Thessaloniki, Greece; 2Department of Oral and Maxillofacial Surgery, University Hospital of Erlangen, 91054 Erlangen, Germany

**Keywords:** bisphosphonates, femoral nerve, immunohistochemistry, NeuN, Sox10

## Abstract

*Background and objectives:* Bisphosphonates represent selective inhibitors of excess osteoblastic bone resorption that characterizes all osteopathies, targeting osteoclasts and their precursors. Their long-term administration in postmenopausal women suffering from osteoporosis has resulted in neural adverse effects. The current study focuses on the research of possible alterations in the femoral nerve, caused by bisphosphonates. We hypothesized that bisphosphonates, taken orally (per os), may produce degenerative changes to the femoral nerve, affecting lower-limb posture and walking neuronal commands. *Materials and Methods:* In order to support our hypothesis, femoral nerve specimens were extracted from ten female 12-month-old Wistar rats given 0.05 milligrams (mg) per kilogram (kg) of body weight (b.w.) per week alendronate per os for 13 weeks and from ten female 12-month-old Wistar rats given normal saline that were used as a control group. Specimens were studied using immunohistochemistry for selected antibodies NeuN (Neuronal Nuclear Protein), a protein located within mature, postmitotic neural nucleus, and cytosol and Sox10 (Sex-determining Region Y (SRY)—High-Motility Group (HMG)—box 10). The latter marker is fundamental for myelination of peripheral nerves. Obtained slides were examined under a light microscope. *Results:* Samples extracted from rats given alendronate were more Sox10 positive compared to samples of the control group, where the marker’s expression was not so intense. Both groups were equally NeuN positive. Our results are in agreement with previous studies conducted under a transmission electron microscope. *Conclusions:* The suggested pathophysiological mechanism linked to histological alterations described above is possibly related to toxic drug effects on Schwann and neuronal cells. Our hypothesis enhances the existing scientific evidence of degenerative changes present on femoral nerve following bisphosphonates administration, indicating a possible relationship between alendronate use and neuronal function.

## 1. Introduction

Bisphosphonates (BPs) are a widely known category of pharmacological agents developed back in the 1880s targeting bone and calcium metabolism disorders [1]. The positive calcium balance achieved by BPs is caused by inhibition of excess osteoclastic bone resorption that characterizes all osteopathies [2].

BPs are known for their effectiveness in osteoporosis treatment, as it has been proved that they contribute to the reduction of bones’ absorption and osteoclasts’ apoptosis. Additionally, BPs that contain nitrogen (Nitrogen-containing Bisphosphonates (NBPs)) are useful in new treatment approaches of neurological disorders. Previous studies have shown that NBPs may have an alleviating effect on Alzheimer’s and Huntington’s diseases [3,4]. Fractures prevention is another reason for choosing BPs as well as advanced skeletal malignant disorders [5].

More specifically, alendronate is used for glucocorticoid-induced osteoporosis’ treatment. This usage is based upon the knowledge that bone mineral density is increased following BP administration [6]. In addition, in patients who have taken glucocorticoids as a treatment for childhood-onset rheumatic disease, early use of alendronate led to prevention of bone’s deconstruction. [7]. Alendronate is a very important pharmaceutical option for postmenopausal osteoporosis treatment showing great improvement of the lumbar spine and femoral neck bone density. Furthermore, following BP administration, a calcium increase in blood serum was observed while phosphorus was decreased [8].

### 1.1. Femoral Nerve

#### 1.1.1. Anatomy and Physiology of the Femoral Nerve

Second, third, and fourth lumbar spinal nerves constitute the femoral nerve, responsible for hip and knee joints, anteromedial thigh skin, and anterior thigh muscles innervation [9,10,11]. Lower-extremity standing is regulated by femoral nerve that innervates the key muscles for knee extension: vastus medialis, vastus lateralis, and vastus intermedius. Rectus femoris not only contributes to the extension of the knee but also flexes the hip. The sartorius has a role in hip and knee flexing, while the pectineus has a limited role in hip flexion. All these functions would be impossible without the electrical stimulation of the femoral nerve [12].

#### 1.1.2. Femoral Nerve Damage

The femoral nerve can be trapped for several reasons: tumors, abscesses, hematomas, and enlarged lymph nodes are just some of the causes that are responsible for femoral nerve compression. Also, nerve tenderness is observed in gynecological surgical procedures in which the hips rotate outward and the thighs abduct. The femoral nerve may be damaged after blocking during anesthesia; nerve pull during surgery; and direct injury resulting in bending, knee extension, delivery, and external hip rotation. Nerve damage can occur after invasive procedures such as hip replacement and knee surgery. Clinically, femoral nerve damage can be revealed by thighs, and physical examination reveals weakness of the quadriceps as well as a reduction or absence of tendon reflexes mainly of the knee [13].

### 1.2. Bisphosphonates

#### 1.2.1. BP Chemistry

BP ((HO)_2_P(O)CR^1^R^2^P(O)(OH)_2_) stable pyrophosphate forms are molecules occurring naturally in the skeletal system, which demonstrate an inhibitory effect on calcification. The substitution of the central oxygen atom by carbon resulted in a phosphorus–carbon–phosphorus (P–C–P) group (Figure 1), thus making them heat and enzymatic splitting resistant, retaining at the same time their crucial bone properties. Biologically active BP derivatives have been synthesized via substitution, gaining characteristic pharmacological properties on bone. Ligands R1 and R2 characterize the various BPs derivatives [1]. The subdivision of BPs into chemical groups is based on side chains’ alphabetic order: first-generation BPs lacking nitrogen substitution (Nonnitrogen-containing Bisphosphonates (NNBPs)), etidronate; second-generation BPs with nitrogen substitution (NBPs) (aminobisphosphonates), pamidronate and alendronate; and third-generation BPs with nitrogen substitution (aminobisphosphonates) and nitrogen atom substitution, ibandronate or aromatic rings containing nitrogen, risedronate with pyridine-rings, and zoledronate containing imidazole-rings (Figure 2) [14].

#### 1.2.2. BP Pharmacodynamics

Although the BPs demonstrate a minimal to low intestinal absorption (<1%–3%) following oral administration, this is outbalanced by their highly augmented potency as even 1% of an administrated dose is effective. Calcium-rich aliments, for instance, dairy products, decrease their absorption when taken together due to the formation of insoluble calcium chelates [15]. BPs’ distribution occurs via blood circulation, after which they accumulate in the bones and their elimination occurs via the renal pathway, escaping metabolism. In the bloodstream, conjugation to albumin takes place, forming bonds with significant differences in strength (from 22% for zoledronate to 87% for ibandronate). Such differences affect BP elimination time from plasma. 

#### 1.2.3. BP Distribution

Throughout circulation, BPs are actively bound to the bone surface, accumulated mainly in the region called resorption lacunae, a sealed zone separating the area between the bone surface and osteoclasts, where they chelate Ca^2+^, targeting the calcium phosphate mineral component of bone, called hydroxyapatite [2]. The accumulation scale depends on the availability of the resorption area and on the affinity for hydroxyapatite crystals of different BPs [1]. The high affinity to bones is connected with the presence of a hydroxyl group in their structure. This group is fixed with calcium ions on the surface of hydroxyapatite [16]. The various mineral-binding affinities of BPs exert a strong effect on their biological potency and duration of action. The nonnitrogenous BPs show weaker binding to hydroxyapatite than nitrogen-containing BPs, which are most widely used because of their double potency [17]. Hence, after BP accumulation at sites of bone resorption, they are selectively encapsulated by active osteoclastic cells.

#### 1.2.4. BP Mechanism of Action

The mechanism of action of BPs occurs in the bone-remodeling process, which consists of a normal and physiological action in which bone formation and resorption occur simultaneously in the same proportion. Regardless of the administration route and dosage regimen, BPs’ inhibition of osteoclastic bone resorption starts 1–2 days after therapy initiation. Targeting osteoclasts and their precursors, BPs interfere with the mevalonate pathway, which is crucial for osteoclastic generation and functionality, by hindering lipid chains of prenylated proteins production, affecting also steroidal metabolism. BPs act in the following steps of mevalonic acid synthesis inhibiting osteoclastic resorption by three different mechanisms, indicating the three different generations of the pharmaceutical agents: (Figure 3) first-generation BPs (nonnitrogen-containing BPs) after being metabolized in the osteoclasts form together with adenosine monophosphate a non-hydrolyzable cytotoxic analog of adenosine triphosphate (ATP), withholding the energy that is needed for isopentenyl pyrophosphate synthesis and inducing osteoclast apoptosis [18,19]. They also may inhibit tyrosine phosphatases in osteoclasts. On the other hand, second-generation BPs that contain nitrogen hinder the enzymatic transformation of dimethylallyl pyrophosphate to geranyl pyrophosphate. Finally, third-generation nitrogen-containing BPs act by preventing the next step of enzymatic reaction transformation of geranyl pyrophosphate to geranylgeranyl or farnesyl pyrophosphate [1]. All nitrogen-containing BPs inhibit farnesyl diphosphate synthase (FPPS) or farnesyl pyrophosphate synthase (FPPS) via binding to the substrate site of the enzyme that plays a key role in the mevalonate pathway, responsible for the production of isoprenoid lipids essential in synthesis of sterols as well as for the posttranslational transformation of GTPase/GTP-binding proteins in a procedure known as protein isoprenylation necessary for osteoclastic functionality [2,19]. Under all circumstances, cells lose certain membrane characteristics crucial for osteoclastic function, provoking this way their apoptosis.

Mechanistically, the first step of the aforementioned procedure includes BPs’ incorporation from the osseous membrane into osteoclastic cells. Following this, BPs induce various modifications that disturb osteoclastic bone resorption, including cytoskeleton structural alterations, microtubule depolymerization as well as actin-ring formation, retraction of the ruffled membrane, and change of bone migration and binding properties. This inhibition in nitrogen-containing BPs increases Isopentenyl Pyrophosphate (IPP) concentration in steroidal and isoprenoid biosynthesis within osteoclastic cells, formatting isopentenyl ATP [1]. BPs inhibit the recruitment and differentiation of osteoclast precursor cells, inducing apoptosis of the differentiated osteoclasts and inhibiting active osteoclasts [17]. Overall, the procedure affects directly osteoclasts by hindering their differentiation or recruitment due to inhibition of proliferation of macrophages that fuse and transform into osteoclastic cells; by reducing osteoclastic activity caused by osteoclast structure alterations especially in cytoskeleton elements that affect bone resorption; by inhibiting osteoclastic attachment to bone matrix; and by causing toxic damage to mature osteoclasts, intensification of osteoclast apoptosis, and in simple terms hindrance of the entire osteoclastic functionality in bone resorption [1,18]. The aforementioned combination leads to caspase recruitment that regulates apoptosis, having a direct effect on the reduction of calcium and bone resorption products in the serum. Apart from their activity on osteoclasts, BPs have a direct effect on osteoblasts, triggering them to generate a factor that has a negative impact on osteoclastic activation and recruitment. Their influence on osteocytes remains mainly obscure [2]. In this manner, BPs ameliorate bone mineral density (BMD) in trabecular bones, an action that is being stabilized to a plateau after years of administration. For instance, it has been shown that alendronate increases in BMD in the proximal femur and reaches a plateau after 3 to 5 years of consecutive administration [20].

#### 1.2.5. BP Adverse and Side Effects

In contrast to the welcoming effects of BPs administration in certain cases, e.g., osteoporosis, the associated symptoms of this particular medical therapy are not to be neglected. The short-term and often more severe long-term side effects of BP treatment are cited below.

##### 1.2.5.1. Short-Term Side Effects

A major effect following intravenous BP administration is hypocalcemia [21], occurring more frequently in hypoparathyroidism, renal failure, calcium and vitamin D insufficiency, and overactivated osteoclastic bone resorption in bone malignancies and Paget disease of bone. Moreover, 10%–30% of the patients receiving BPs for the first time appear to experience an acute inflammatory response, including arthralgia, headache, and influenza-like symptoms [22]. The suggested mechanism of this response focuses on pro-inflammatory peripheral blood gamma-delta (γδ) T cells activity, resulting in cytokine production. Rarely reported cases are of ocular inflammation, ocular painful conditions, and photophobia with both oral and intravenous (IV) BPs contributing to these conditions [21]. Musculoskeletal painful conditions could be present following the administration of all oral and IV BP preparations. However, according to the Food and Drug Administration (FDA), serious musculoskeletal pain is closely linked with BP administration. Other potential side effects of oral BP administration are esophageal irritation and erosion, possibly resulting in esophageal cancer. These complications affect mainly patients with esophageal pathology including stricture and reflux. After all, there is still a need for further data examining the correlation between BP administration and the causes of upper gastrointestinal adverse effects [21,22].

##### 1.2.5.2. Long-Term Side Effects

###### Osteonecrosis of the Jaw (ONJ)

No other concern among those regarding BP treatment has gained a greater interest in the last years than Osteonecrosis of the Jaw (ONJ) [22]. Above all, patients with oncologic problems in treatment with high doses of IV BPs, more specifically with zoledronic acid and pamidronate, developed ONJ in 94% of all cases. Patients suffering from cancer undergoing oral BP treatment have a frequency of developing ONJ 1%–10% whereas, in non-cancer patients, seems to be 1 in 10,000 to 1 in 100,000 patients, although these estimations are merely based on uncompleted data. Other ONJ risk factors include inappropriate oral cavity hygiene, dental surgery, and denture placement [21]. As a result, a causal relationship between the use of BPs and ONJ development is yet to be established [22].

###### Atrial Fibrillation

Atrial fibrillation (AF) is another concerning side effect of BPs’ prescription, although there is no convincing mechanism and correlation between the prescribed dose and the duration of BPs treatment. Increased prevalence of AF is not established by other conducted studies [21,22].

###### Bone Turnover Suppression and Atypical Femoral Fractures (AFFs)

There are reports made in 2005 by Odvina et al. suggesting an over suppression of bone remodeling caused by BPs therapy, complicating bone microfracture cure and augmenting bone fragility [21]. This suggestion is based on BPs’ mechanism of action in inhibiting osteoclast activity, thus producing a “frozen bone”, as described above [22]. However, a generalization of case reports is impossible since other components might be of great importance in the suppression of bone turnover as well, such as exposure to varying doses of BPs, reception of other pharmaceutical agents that have an action on bone for instance glucocorticoids, as well as underdiagnosed bone pathology before BP therapy initiation [21,23]. Several reviews summarize the background of Atypical Femoral Fractures (AFFs): the majority of patients had been exposed to BPs (93.9%), and they often used the treatment for osteoporotic bones (92.3%), whereas a smaller number of patients used BPs for malignant conditions being mainly of smaller age and female sex contrary to subjects with femoral disorders caused by osteoporosis. Despite this work, there is still much uncertainty regarding the pathogenesis of these fractures and whether they are actually BP related or coincidentally linked to former BP exposure [21,24]. AFFs are partly what we were searching for by using immunohistological markers on the femoral bone.

## 2. Materials and Methods

### 2.1. Animals

The experimental part included twenty female 12-month-old Wistar rats having a weight of 500 g and kept separately in cages. They were under controlled temperature and humidity conditions and light-dark cycle lasting half a day. Two animal groups were formed; 10 animals represented the experimental group A and an equal number of animals was the control group B. The chosen BP (alendronate; Fosamax^®^, Merck & Co., Inc., New Jersey, NY, USA) was given per os (po) at a dose of 0.05 mg/kg body weight/week for thirteen consecutive weeks to group A. The chosen dose was based on the one normally given to human patients. The drug was administered thirty minutes before breakfast once a week for 13 weeks. Group B was given po normal/physiological saline as a placebo. After euthanasia, the femoral nerve of the animals was removed and specimens were processed for immunohistochemistry and optical microscopy examination. The Aristotle University Medical School Bioethics Committee consented to the overall research and approved this study with ethical approval number 1.135 (approval date 8/11/12).

### 2.2. Dose

BPs are administered to humans at a dose of 1 mg/kg/week. Up to date, alendronate metabolism in animals has not been proved. Taking into consideration that the oral bioavailability of the drug in humans is 0.76%, the dose adaptation in animals was calculated to 0.05 mg/kg/week to simulate human organism pharmacokinetics, avoiding at the same time the toxic side effects. The decision was also dictated by the fact that per os dose of 0.05 mg/kg has been prior utilized in animal research [25]. After all, variations in drug metabolism causing different results between animals and humans are naturally expected.

### 2.3. Hematoxylin-Eosin staining

In order to perform the hematoxylin-eosin staining, the femoral nerve tissue samples were firstly dehydrated in escalating 25%, 50%, 70%, 80%, and 96% v/v and absolute alcohol, cleared with xylene and fixed in paraffin. Thin cuts of 5 μm were made, and after deparaffinization, they were stained with hematoxylin-eosin and covered with mounting medium and coverslips. The observation was made under a Zeiss microscope and the photography with a Contax camera. Compared to the control group, the experimental group showed a possible thickening of the myelin sheath and/ or neuronal axons which was necessary to be examined further employing immunohistochemistry (Figure 4).

### 2.4. Immunohistochemistry

Firstly, the pieces of femoral nerve were incorporated in paraffin, after which thin cuts of 3–4 μm were made. The Polymer Detection System Kit was employed for the procession of the sections, following the dictated guidelines. Deparaffinization in xylene was the first step. Afterwards, specimens were immersed in digressive densities of 100%, 96%, and 70% v/v of absolute ethanol. Distilled water wash was the following step. Incubation at 65 °C was essential in order to achieve the antigen retrieval. Then, specimens were washed with Phosphate-Buffered Saline (PBS) and were incubated in hydrogen peroxide for 5 minutes to block endogenous peroxidase activity. They were then rinsed for a second time with PBS buffer. Primary antibody solution coverage was the next step. The two primary antibodies used were the NeuN (Feminizing Locus on X-3, Fox-3, Hexaribonucleotide Binding Protein-3, Abcam Inc., Cambridge, MA, USA) and the SOX-10 (SRY-related HMG-box, Abcam Inc., Cambridge, MA, USA) already set and steady to be used. Finally, they were washed using deionized water. To detect the immunohistochemical staining, specimens were immersed in Post-Primary solution, were washed, and then were immersed in a polymer solution and Chromogen-diaminobenzidine solution. The final stage was Hematoxylin staining. Specimens were washed in water and processed with escalating 70%, 96%, and 100% v/v densities of ethanol and xylene solution in order to remove humidity. Each of the 2 antibodies that were examined has undergone the abovementioned immunohistochemical staining procedure. A positive control tissue was run during the staining procedure with brain tissue for the NeuN antibody and skin tissue for the Sox10 antibody respectively. For the negative control, each of the two primary antibodies was replaced with PBS buffer. For the specimens’ optical observation, a Zeiss microscope was employed following by digital photography utilizing a Contax camera.

### 2.5. Immunohistochemical Markers

#### 2.5.1. NeuN

NeuN is a protein abundant in central and peripheral nervous system neuronal cell types. The NeuN protein was first discovered in 1992 and is exclusively detected in the nervous tissue, more specifically in the nuclei and perinuclear cytoplasm of mature, postmitotic neurons that are terminally differentiated and are no longer able to divide. In the nucleus, NeuN is primarily located in euchromatin areas; however, its binding properties regarding the RNA are seemingly of greater importance than those regarding the DNA. NeuN’s unique properties (its expression only in mature cells of the nervous tissue and especially in their nucleus) provide immunohistology with an extremely selective neuromarker that binds A60 antibodies, thus shedding light in scientific research and histopathological diagnosis (e.g., identification of neuronal differentiation or changes in existing neuronal populations and application in neuro-oncology) [26].

#### 2.5.2. Sox10

Myelin is a specialized multilayer membrane essential for the rapid axonal transmission and for axonal trophic support that averts axonal degeneration. The peripheral nervous system’s Schwann cells and the central nervous system’s oligodendrocytes are responsible for myelin production. The transcription factor Sox10 is pivotal for the myelin regulation in both cell types [27,28,29,30]. SOX10 transcription factor has a name that illustrates the similarity to the Sex-determining Region Y (SRY), High-Motility Group (HMG)-box on the Y-chromosome and is abbreviated from SRY-related HMG-box. In other words, it is an HMG-domain-containing transcription factor that plays a key role when Schwann cells specify and develop, maintaining peripheral nerve myelination [28,30,31].

Melanocytes and Schwann cells primarily express Sox10 as they both originate from the neural crest [30,32]. In particular, Schwann cells (SCs) originate from neural crest cells that firstly produce SC precursors and then immature SCs, pro-myelinating SCs, and finally non-myelinating and myelinating SCs [33]. Additionally, it has been proved that Sox10 is also expressed in myoepithelial and basal gland cells [30,34]. Melanocytes and Schwann cells’ development and maintenance are mainly subjected to Sox10 [30]. As far as myelination is concerned, every single step of Schwann cells’ development necessitates Sox10, which provokes the activation of certain transcription factors that cooperate with this factor in the following developmental stages. Mechanistically, at the immature stage, expression of Sox10 and Histone deacetylase (Hdac1/2) epigenetic regulators is essential for progression to the promyelinating stage [27,31,35]. Oct6 is a Sox10 target in pro-myelinating Schwann cells and binds with the transcription factor, activating the zinc finger transcription factor Egr2/Krox20, another critical determinant of myelination that is required for myelin formation and maintenance. Terminal differentiation is induced after this step, making the linkage between Sox10 and Krox20 crucial for myelination in the peripheral nervous system [28,29]. More specifically, Sox10 activates with the collaboration of Egr2 myelin gene transcription joining distal and intronic enhancer sites in peripheral myelin genes such as the myelin basic protein myelin protein zero [28,33]. Sox10 is essential for the specification of Schwann cells from the neural crest, while Egr2 expression is activated by axonal signals in myelinating Schwann cells and is required for cell cycle arrest and myelin generation [27]. As a result, the loss of Sox10 causes equal impairment in Egr2 expression as well as the degradation of the myelin sheath that result in axonal catastrophe and impaired nerve conduction [33].

When myelin structure and functionality are being modified, various peripheral nerve disorders can occur, such as Charcot-Marie-Tooth disease leading to persistent polyneuropathy [28,36]. Sox10 function impairment can cause type IV Waardenburg syndrome with sensorineural hearing loss, oral and hair pigment systems defects, and oral lateral displacement being the main signs and, finally, can increase the risk for Hirschprung disease genetic variant affecting the intestinal autonomic nervous system [30]. The aftermath of nerve injury is the induction of Schwann cells to become unmyelinated but, at the same time, to produce agents that help nerves to regenerate [28].

Based on the aforementioned statements, Sox10 has been exploited as an immunohistochemical marker for damages in Schwann cells and melanocytes as well as dermal and soft tissue tumors [30]. Anti-Sox10 antibody has priorly been applied as a sensitive and specific tracer to a variety of normal tissues as well as malignancies that originate from neural crest and tumors of mesenchymal and epithelial origin, making Sox10 a Schwannian and melanocytic marker of great importance for its significant sensitivity and specificity in an effort to diagnose neural crest-derived anomalies [34,35,37].

## 3. Results

The intensity of staining was evaluated employing an optical microscope with a qualitative method as negative (−), week (+), moderate (++), and strong (+++) by 3 different reviewers. The control and experimental groups were equally NeuN positive (++). Immunohistochemical staining for Sox10 showed that samples extracted from rats given alendronate were more positive (++) compared to samples of the control group, where the marker’s expression was not so intense (+). Our results are in agreement with a previous study conducted under transmission electron microscope, suggested ultrastructural alterations of the femoral nerve [38], and confirm that degenerative changes in the myelin sheath occur following BP administration. Study of the inferior alveolar nerve under the same conditions evidenced myelin vacuolization, axonal detachment, local thickening, and/or myelin disruption (Figure 5) [15].

## 4. Discussion

Generally speaking, nerve degeneration is histologically indicated by axonal swelling, axoplasmic darkening, axonal detachment, thickening and decrease, demyelination, myelin disruption, and vacuolization of the myelin sheath [39]. The present study proposes that BP administration could result in ultrastructural alterations in the myelin sheath, possibly involving one or more of the above mechanisms, while the microstructure of the axons may probably remain unaltered.

The pathophysiological mechanism behind myelin sheath damage following therapy with BPs remains obscure. A hypothesis can be made that Schwann cells’ ischemia may result in the observed degenerations. Similar damage, for instance, vacuolization of Schwann cells (and mostly of satellite cells), was observed in the spinal ganglia of dogs following aortic occlusion, evidence that proves that ischemic impact to Schwann cells can cause the aforementioned damage [40]. Direct metabolic or toxic changes on the neuron resulting in hindering in vitro proliferation may be a possibility, as Schwann cells have a strong connection with myelin synthesis, strengthening the hypothesis that modifications to Schwann cells are strongly related to mental nerve neuropathy after BP administration [41].

It is generally recognized that the myelin sheath is susceptible to ischemia. Recent studies have proven demyelination and ultrastructural changes of myelin sheath seven days after brain ischemia [42]. Schwann cells are a well-known target in ischemia-reperfusion injury of peripheral nerves, as Schwann cell apoptosis occurs during reperfusion and persists during axonal regeneration, leading to impairment of axonal function and fiber regeneration efficiency [43].

It was shown in the past that chronic endoneurial ischemia could cause peripheral neuropathy; sciatic nerve reduction in endoneurial blood flow resulted in structural abnormalities at nodes of Ranvier and mild axonal atrophy, while segmental demyelination and degeneration of axons were not involved [44]. More recently, the study of optic nerve ischemia associated with non-arteritic anterior ischemic optic neuropathy showed axon and myelin structural alterations [45].

A reversion of these effects, providing neuroprotection and remyelination, is considered the main therapeutic goal, aiming at the restoration of the function of demyelinated axons. Taking into consideration that certain anomalies are characterized by myelin sheath abnormalities but not by demyelination, the application of remyelinating strategies to these disorders require previous knowledge of whether the myelin damage results from an initial damage in myelin or the oligodendrocytes or both and whether the aforementioned damage results in myelin damage and demyelination [46].

## 5. Conclusions

In the framework of this study, we provided evidence that BP administration may lead to alterations of the femoral nerve, affecting the rigidity of myelin sheath in the experimental group in comparison with the control group.

It must be taken into consideration that our results may have been affected by the selected animals and the chosen dose; a higher dose may have resulted in more intense alterations. Moreover, the pharmacokinetics and the duration and route of administration could have undoubtedly been a major influence on the outcome. Furthermore, the current study addressing only morphological changes makes the neurophysiological assessment of the nerve examined essential for further investigation.

These outcomes may be underlined by forthcoming investigations leading to analogous nerve degenerative changes not only in femoral but also in other nerves, as shown previously with the inferior alveolar nerve [15]. Disambiguation may include studies in human cadavers of patients undergoing BP treatment with the purpose of shedding light upon the pathophysiological mechanism of nerve degeneration, the reversibility of the changes after the end of BP therapy, and the probability of the future reevaluation of BP therapeutic regimen.

## Figures and Tables

**Figure 1 medicina-56-00140-f001:**
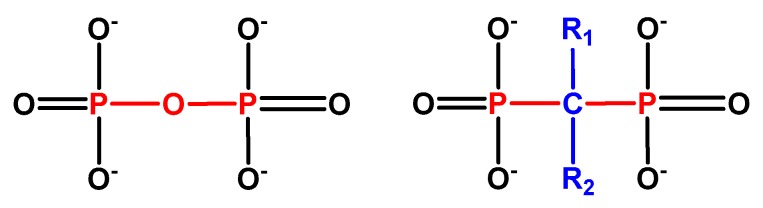
Pyrophosphate/bisphosphonates chemical formulas.

**Figure 2 medicina-56-00140-f002:**
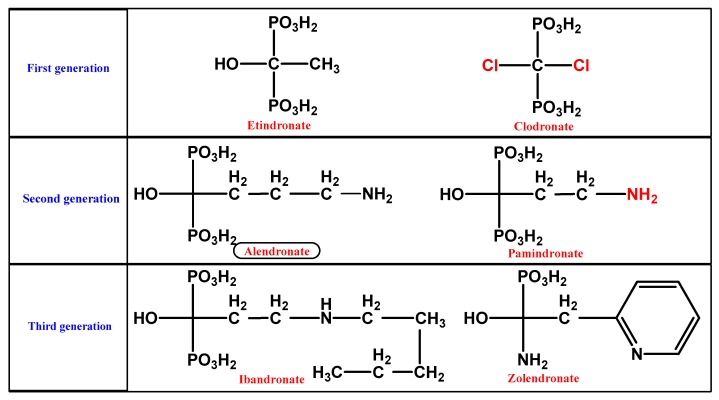
The three generations of bisphosphonates.

**Figure 3 medicina-56-00140-f003:**
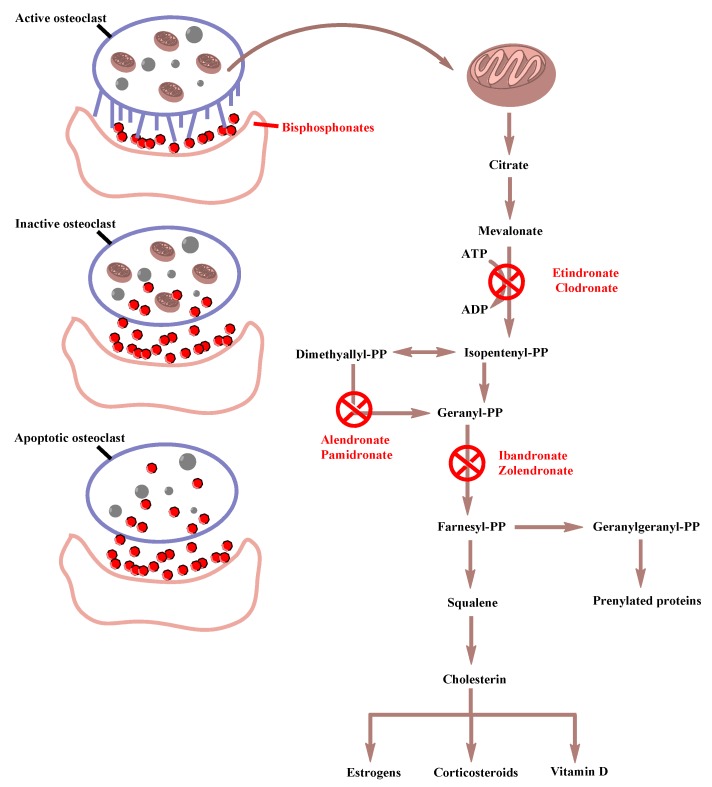
Mechanisms of action of the three generations of bisphosphonates.

**Figure 4 medicina-56-00140-f004:**
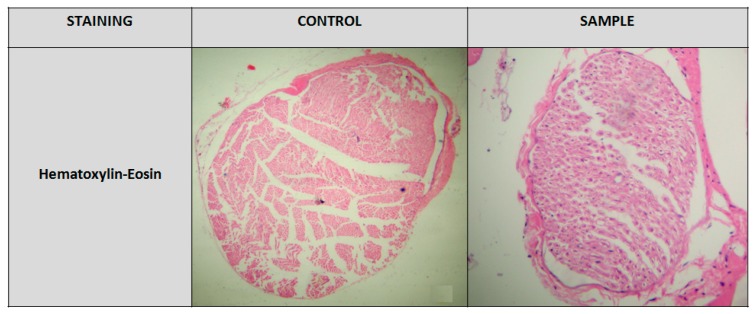
Hematoxylin-eosin staining: Experimental group showed possible myelin and axonal thickening compared to the control group. (Focus: 10 × 4).

**Figure 5 medicina-56-00140-f005:**
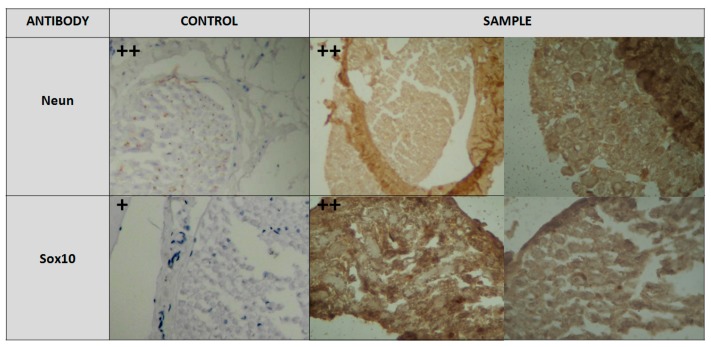
Immunohistochemistry: Neun staining: control and experimental groups were equally NeuN positive (++); Sox10 staining: experimental group was more Sox10 positive (++) compared to control group (+). (Focus: control 4 × 10, sample 4 × 10 and 40 × 7.7).

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
