# Peer review of "Immunohistochemical Femoral Nerve Study Following Bisphosphonates Administration"

_medicina, 2020, doi:10.3390/medicina56030140_

Round 1

Reviewer 1 Report

The present study aims to examine the influence of bisphosphonate on nerves in an experimental model. The authors found a possible relationship between alendronate use and neuronal function. Generally speaking, the study is well designed. I would like to recommend its publication as long as the authors follow my suggestion. First, there are several abbreviations in the abstract without providing their original expressions, such as os. Please provide their original abbreviation. Second, I think the introduction might be too lengthy. I would like the authors to focus more on the literature regarding the neuro-degenerative effect of Bisphosphonate. Third, in line 55, please cite the following references to address the anatomy of the femoral nerve and its related branches: (1) Sonographic Tracking of the Lower Limb Peripheral Nerves: A Pictorial Essay and Video Demonstration. Am J Phys Med Rehabil. 2016 (2) Ultrasound Imaging for the Cutaneous Nerves of the Extremities and Relevant Entrapment Syndromes: From Anatomy to Clinical Implications. J Clin Med. 2018 Fourth, in line 216, is there any intervention applied on the control group, like placebo injection? Please clarify.

Author Response

Dear Mr. / Ms. Reviewer,

Thank you for giving me the opportunity to submit a revised draft of my manuscript titled “Immunohistochemical femoral nerve study following bisphosphonates administration” to Medicina. We appreciate the time and effort that you have dedicated to providing your valuable feedback on our manuscript. We are grateful for your insightful comments on our paper. We have been able to incorporate changes to reflect your suggestions. We have highlighted the changes in yellow within the manuscript sent to you as Word. Here is a point-by-point response to your comments and concerns.

Point 1: First, there are several abbreviations in the abstract without providing their original expressions, such as os. Please provide their original abbreviation.

Response 1: Thank you for pointing this out. We agree with this comment. Therefore, we have added the original expressions/ explanations in lines 15, 40, 75, 81, 125, 144, 173, 175, 177, 186, 207, 220, 241, 272, 284

Point 2: Second, I think the introduction might be too lengthy. I would like the authors to focus more on the literature regarding the neuro-degenerative effect of Bisphosphonates

Response 2: We appreciate your comment on the introduction’s length in which we wanted to incorporate shortly and effectively all the major facts concerning Bisphosphonates, essential for the analysis of parts that follow. We would also like to add more information regarding the neuro-degenerative effect of Bisphosphonates, but due to the fact that this adverse action was not studied in the past, we chose to focus on the most important side effects of these pharmaceutical agents, as seen in the literature.

Point 3: Third, in line 55, please cite the following references to address the anatomy of the femoral nerve and its related branches

Response 3: We have accordingly done this addition. The references in the main text will be renumbered when review process will reach to an end.

Comment 4: Fourth, in line 216, is there any intervention applied to the control group, like placebo injection? Please clarify

Response 4: Thank you for pointing out this important omission. We have incorporated the intervention in line 223.

Please see the relevant attachment.

Sincerely,

Karakousis Vasileios Alexandros

Reviewer 2 Report

  • In the section “Introduction”, Authors affirm that several symptoms in Alzheimer and Huntington diseases can be moderated with NBPs administration: They should provide some references.
  • In the section “Materials and Methods” at paragraph “2.3. Immunohistochemistry”, Authors should correct the description of the final staining (which is Hematoxylin staining, not Hematoxylin-eosin).
  • In the section “Results”, Authors cite an article that describe ultrastructural alterations of the inferior alveolar nerve in wistar rats after alendronate administration (Dietrich et al.). Have Authors analyzed ultrastructural alterations? Furthermore, have they found histological differences between control and treated group? It could be interesting to see a Hematoxylin-Eosin section of the two different groups.
  • In the same section, Authors gave a qualitative evaluation of the staining. Have They noted any quantitative differences of immunostaining?
  • In the same section, there isn't a positive and a negative control of the antibody. It shall be provided.
  • In the section “Conclusions”, Authors affirm “we provided evidence that BPs administration may lead to ultrastructural alterations of the femoral nerve”. There is not a direct evidence of this alteration, it’s only an hypothesis based on an indirect effect evaluating by immunoistochemestry. It’s premature to talk about ultrastructural or structural changes, this assertion is not substantiated by any evident demonstration.
  • It could be interesting to see the effects of the same treatment in vitro. Have Authors made any study in vitro?

Author Response

Dear Mr. / Ms. Reviewer,

Thank you for giving us the opportunity to submit a revised draft of my manuscript titled “Immunohistochemical femoral nerve study following bisphosphonates administration” to Medicina. We appreciate the time and effort that you have dedicated to providing your valuable feedback on our manuscript. We are grateful for your insightful comments on our paper. We have been able to incorporate changes to reflect your suggestions. We have highlighted the changes in light blue within the manuscript sent to you as word. Here is a point-by-point response to your comments and concerns.

Point 1: In the section Introduction, Authors affirm that several symptoms in Alzheimer and Huntington diseases can be moderated with NBPs administration. They should provide some references.

Response 1: Thank you for pointing out this important omission. We have included the necessary references concerning this matter in line 42, making at the same time slight changes to the sentence’s expression. The references in the main text will be renumbered when the review process will reach to an end.

Point 2: In the section Materials and Methods at paragraph 2.3. Immunohistochemistry, Authors should correct the description of the final staining, which is Hematoxylin staining, not Hematoxylin-Eosin

Response 2: We appreciate this correction. The staining has been changed to Hematoxylin

Point 3: In the section Results, Authors cite an article that describes ultrastructural alterations of the inferior alveolar nerve in Wistar rats after alendronate administration (Dietrich et al.). Have Authors analyzed ultrastructural alterations? Furthermore, have they found histological differences between control and treated group? It could be interesting to see a Hematoxylin-Eosin section of the two different groups.

Response 3: We have studied the ultrastructural alterations of the femoral nerve following Bisphosphonates administration in another study, as it is being mentioned in line 313. At the time of the document’s writing, the beforementioned study had not been published. As it was published recently, we added the corresponding reference in line 313. As far as the Hematoxylin-Eosin staining, we agree that this addition would help to clarify the whole matter. Thank you for your suggestion. The addition of our observations as well as the additional table has been made before the section immunohistochemistry.

Point 4: In the same section, Authors gave a qualitative evaluation of the staining. Have They noted any quantitative differences of immunostaining?

Response 4: The aim of the current study was to observe, if any, qualitative differences between the control and experimental group after bisphosphonates administration. Quantitative analysis may be a part of a future immunohistochemical analysis of the femoral nerve. Thank you for this suggestion though.  

Point 5: In the same section, there isn’t a positive and a negative control of the antibody. It shall be provided

Response 5: You have raised an important point here. We have accordingly added the controls in lines 284 and 328 respectively.  

Point 6: In the section Conclusions, Authors affirm ‘’we provided evidence that BPs administration may lead to ultrastructural alterations of the femoral nerve’’. There is not direct evidence of this alteration, it’s only a hypothesis based on an indirect effect evaluating by immunohistochemistry. It’s premature to talk about ultrastructural or structural changes; this assertion is not substantiated by any evident demonstration.

Response 6: We agree with the above statement, thus we modified the expression of the sentence in line 379. 

Point 7: It could be interesting to see the effects of the same treatment in vitro. Have Authors made any study in vitro?

Response 7: Thank you once again for your suggestion. It would have been interesting to explore this aspect, in order to review the topic in a more spherical way. In vitro studies of the effects of bisphosphonates on several tissues might be a future aim of our scientific work.

Please see the relevant attachment.

Sincerely,

Karakousis Vasileios Alexandros

Round 2

Reviewer 2 Report

  1. Related to previous point 5. “…there isn’t a positive and a negative control of the antibody. It shall be provided…”

I mean a positive and negative control of the immunostain analysis carried by Authors.

For positive control: Authors should show the immunostain of Neu-N and SOX10 in tissue sample that normally expressed these proteins, to compare the results of the staining in control and treated samples.

For negative control: To verify the specificity of the immunoistochemical stain in control and treated samples, the researcher have do a double stain: one with the primary antibody and one without the primary antibody. In this way Authors prove the specificity of the antibody (for example a non-specific result can be due to DAB reaction).

  1. Related to previous point 6. I did that comment because in the first version of the paper, Authors did not provided the result of Their previous study. Now the is an evidence of ultrastructural alterations, so They could affirm this conclusion.

Author Response

Dear Mr. / Ms. Reviewer,

We have highlighted the new changes in light blue within the resubmitted manuscript. Here is a point-by-point response to your comments and concerns;

Point 1: Related to previous point 5: …there isn’t a positive and a negative control of the antibody. It shall be provided… I mean a positive and negative control of the immunostain analysis carried by Authors. For positive control: Authors should show the immunostain of NeuN and Sox10 in tissue sample that normally expressed these proteins, to compare the results of the staining in control and treated samples. For negative control: to verify the specificity of the immunohistochemical stain in control and treated samples, the researcher have to do a double stain; one with the primary antibody and one without the primary antibody. In this way Authors prove the specificity of the antibody (for example a non-specific result can be due to DAB reaction). 

Response 1: The positive and negative controls are being mentioned more precisely in line 266.  

Point 2: Related to previous point 6. I did that comment because in the first version of the paper, Authors did not provide the result of Their previous study. Now there is an evidence of ultrastructural alterations so They could affirm this conclusion.  

Response 2: The word hypothesis has been changed with the word evidence in line 376.

Sincerely,

The Authors